



**Study of the effect of local forcing on the fractal behavior of**
**shallow groundwater levels in a riparian aquifer**
Abrar Habib[1], Athanasios Paschalis[2], Adrian P. Butler[2], Christian Onof[2], John P. Bloomfield[3], James P. R.
Sorensen[3]
[1] Civil Engineeing Department, University of Bahrain, Kingdom of Bahrain
[2] Department of Civil and Environmental Engineering, Imperial College London, London, SW7 2AZ, UK
[3] British Geological Survey, Maclean Building, Crowmarsh Gifford, Wallingford, Oxon, OX10
*Correspondence to:* Abrar Habib (abr.habib@gmail.com / amhabib@uob.edu.bh )
**Abstract.** With the help of a physically based recharge-groundwater flow model and robust detrended fluctuation
analysis (r-DFAn), the effect of local (catchment-scale) forcing on groundwater levels' scaling behavior in a
riparian aquifer in Wallingford, UK, is investigated. The local forcings investigated in this study include the
rainfall's temporal scaling behavior (which is simulated by changing rainfall's intermittency parameter in a $\beta$ -
lognormal multiplicative random cascade model), the aquifer's physical parameters (saturated hydraulic
conductivity, specific yield, the empirical coefficients of the water retention curve, and the river stage's scaling
behavior).
Groundwater level's scaling behaviour was found to be most sensitive to rainfall's fractal behaviour. Additionally,
there is preliminary evidence suggesting that changes to the rainfall's local scaling behaviour (i.e., change to the
series' scaling that induces crossovers) affects the groundwater's and the recharge's local scaling behaviour.

**1 Introduction**
Fractal behaviour of a time series indicates how the time series statistics depend on scale, and has various
implications. A major implication in water resources management is the level of persistence of a series, i.e., its
likelihood to remain at its current value (Williams & Pelletier, 2015). Depending on the time series, implications
of this may vary. In water resources management, the likelihood of a variable to remain at a high or a low value
is certainly of significance when studying flood risks or planning for potential dry periods or droughts (Habib, A.,
29  2020).
In the field of hydrology, the fractal behaviour of hydrological time series has long been acknowledged
(Kantelhardt et al., 2001; Li & Zhang, 2007; Little & Bloomfield, 2010; Matsoukas, Islam & Rodriguez-Iturbe,
2000). The fractal behaviour of a hydrological time series is a 'fundamental hidden order' (National Research
Council, 1991), i.e. a property that is inherent in hydrological time series that can be quantified but not necessarily
visually noticed. Being able to simulate this 'fundamental hidden order' and study the factors that affect it helps
in gaining insights into the processes and variables being simulated. It is for this reason, among others, that
researchers have gained interest in modelling fractal behaviour and studying it. Various researchers have modelled
fractal behaviour of hydrological and other variables by converting simple and known models from the time-space
domains to the spectral domain (Table A. 1), and others, more recently, used physically-based models in the time
domain to simulate hydrological (or related) variables while incorporating fractal behaviour of the system being
modelled by analysing the outputs and/or inputs using various known techniques (with power spectral analysis
being the most commonly used). This helped them gain insights into the variables/processes being modelled
(Table A. 1).
To present a general picture of previous efforts for using models to incorporate or simulate fractal behaviour, a
non-exhaustive list is presented in chronological order in Appendix A (Table A. 1). Spectral analysis was found
to be the method of preference for studying fractal behaviour of time series by most researchers (Table A. 1),
weather for representing the entire hydrological process in the frequency domain, like in (Gelhar, 1974), or simply
for analysing the input and/or output time series with Fourier transform.
In this work a physically based model is used to study the fractal behaviour of groundwater levels. However, the
novelty of this work lies, firstly, in the use of robust detrended fluctuation analysis, r-DFAn (Habib, A. et al.,
2017), to objectively study the fractal behaviour of groundwater levels and the fractal behaviour of the input
forcing. This enables reliable comparison between various series, and it enables the systematic study of changes
to the scaling regime (which will be referred to as 'local scaling behaviour'). Secondly, rainfall series of varying
fractal properties are simulated and used to drive the physically based model and the benefits of this are addressed
in the relevant section below.





The following section is the Methodology Section which explains the procedure adopted to study the sensitivity
of simulated groundwater levels' fractal behaviour to the various inputs and parameters required to run the coupled
recharge-groundwater flow model. The stochastic rainfall model used to simulate rainfall series of varying fractal
behaviour is also detailed in that section. The section following that is the Results and Discussion Section that
presents the results in two parts, the first includes that forcing that produced statistically significant groundwater
level fractal behaviour and the second includes those that didn't. This is followed by the Conclusion Section.

**2 Methodology**
**2.1 Study Site**
The study site is located in Wallingford, United Kingdom (Figure 1), and it comprises a shallow riparian aquifer,
of about 5m depth, with groundwater levels that exhibit fluctuation over a wide variety of time scales. The data
monitored at Wallingford includes high resolution 1-minute groundwater levels and river stage, 15-minute
rainfall, among other meteorological variables, all of which are summarized in Table 1 and the gauge locations
are indicated in Figure 1. The data available are 4 years long spanning from January 2012 to January 2016.

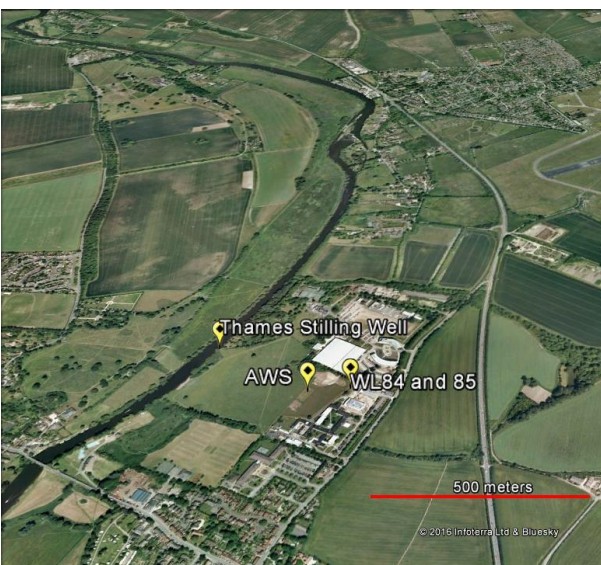

**Figure 1: A Google Earth image of the study site in Wallingford, United Kingdom, with the automatic weather**
**station (AWS), the Thames stilling well and the groundwater boreholes (WL84 and 85) indicated © Google Earth**
**2022.**
**Table 1: Summary of the data measured at the study site from January 2012 – January 2016**

| Datasets | Measuring Station | Time Resolution (minutes) |
|---|---|---|
| **Meteorological Data** | | |
| Rainfall | Automatic weather station (AWS) | 15 |
| Dry bulb temperature | | 15 |
| Wet bulb temperature | | 15 |
| Net solar radiation | | 15 |
| Wind speed | | 15 |
| **Hydrological Data** | | |
| Groundwater levels | WL84 | 1 |
| River Stage | Thames stilling well | 1 |




## 2.2 Fractal Behavior Quantification

The data have been analysed for fractal behaviour using robust detrended fluctuation analysis (r-DFAn) (Habib, A. et al., 2017). r-DFAn is a recently developed procedure that utilizes the well-known detrended fluctuation analysis (Peng et al., 1994) and a number of statistical models to estimate reliable scaling behaviour. The statistical models used were robust regression, to estimate a global scaling exponent as explained in Figure 2, piecewise linear regression to estimate optimum crossover locations, analysis of covariance (ANCOVA) to determine whether the local scaling exponents (explained in Figure 2) were statistically different or not, and multiple comparison procedure to enable comparing three or more groups of data, which is the case when having three or more local scaling exponents. A detailed explanation of r-DFAn can be found in (Habib, A. et al., 2017).

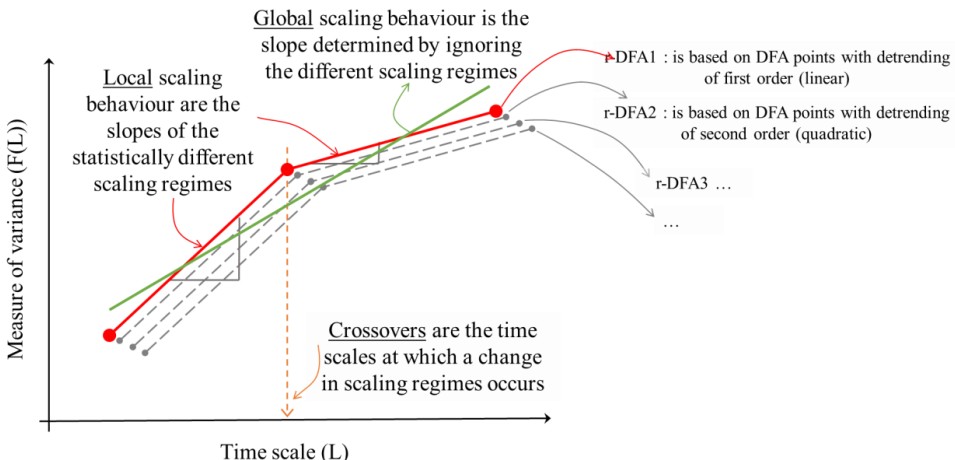

**Figure 2. Explanation of the various components of fractal behaviour that robust detrended fluctuation analysis (r-DFAn) quantifies.**

## 2.3 Groundwater Levels Simulation

Groundwater levels at the study-site are simulated using a recharge-groundwater flow model (Habib, Abrar et al., 2022). The model comprises of a Soil Moisture Accounting Procedure (SMAP) to simulate recharge (Mathias et al., 2015), and a 1D non-linear partial differential equation (the Boussinesq Equation) to simulate groundwater levels with a no-flow boundary at one end and a time-varying specified head boundary at the River Thames. The model is written in MATLAB with explicit discretization for the SMAP, which is derived from a simple water balance integrated over the depth of a soil column (Mathias et al., 2015), and implicit discretization of the Boussinesq Equation (Habib, Abrar et al., 2022). Potential evapotranspiration is estimated from the meteorological data monitored in Wallingford using the procedure explained in FAO Irrigation and Drainage Paper 56 (Allen, et al, 1998). A total of 14 parameters are included in the sensitivity analysis. The sensitivity analysis is performed using Latin Hypercube sampling which involved a total of 12,000 model runs. As a result, 6 parameters are identified as sensitive (showed in Figure 3). Multi-objective optimization using a pattern search algorithm (Custódio et al., 2011) is used to determine the non-dominated parameter sets of the sensitive parameters. A total of 21 unique non-dominated parameter sets are identified. A mathematical representation of the model, the sensitivity analysis and optimization are presented in detail in (Habib, Abrar et al., 2022). A summary of the working of the model along with the input series and sensitive parameters is presented in Figure 3. The model runs at a spatial resolution of 5m and a temporal resolution of 15 minutes.





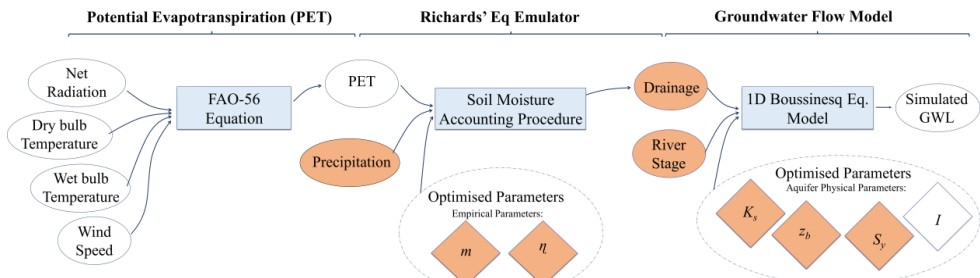

**Figure 3. A schematic showing the input time series and sensitive parameters of the recharge-groundwater flow**
**model. Ovals represent time series, diamond shapes represent sensitive parameters and rectangles represent an**
**algorithm. Orange highlighted shapes are the parameters/time series that are involved in the sensitivity study of**
**groundwater levels' fractal behaviour. PET is potential evapotranspiration, m and η are empirical parameters from**
**the recharge model (one value for summer and one for winter for each parameter), $K_s$ is the hydraulic conductivity of**
**the saturated zone, $z_b$ is the elevation of the base of the aquifer from ordnance datum, $S_y$ is the specific yield of the**
**aquifer, and I is the constant inflow near the no-flow boundary.**
The non-dominated (i.e. optimum) groundwater level simulations are presented in Figure 4, however, for the
purpose of this research, one of these time series will be selected based on its performance in the fractal domain
(i.e. its r-DFAn results). The selected GWL simulation is shown in Figure 4 and its fractal behaviour is presented
in Figure 5. The selected simulation has a Nash Sutcliff Efficiency of 0.716.

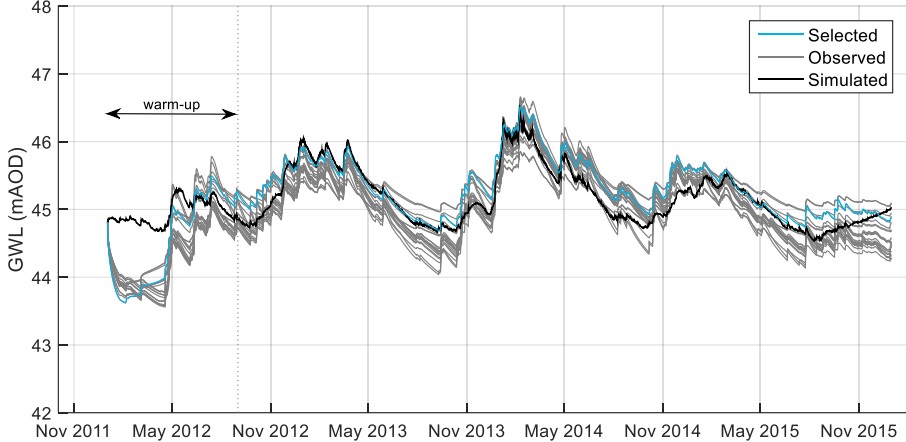

**Figure 4. Simulated non-dominated groundwater levels (GWL) using the coupled recharge-groundwater flow model**
**and observed GWL.**



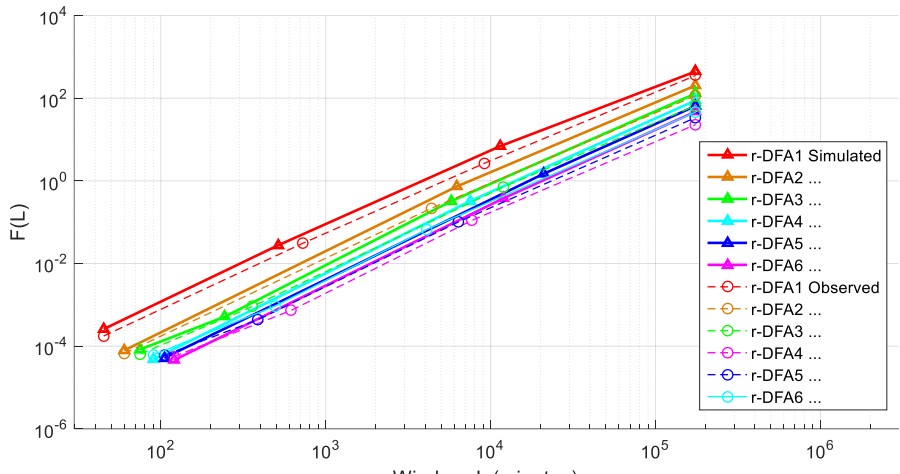

**Figure 5. Fractal behaviour of the selected groundwater levels simulation and how it compares to the fractal**
**behaviour of observed groundwater levels.**
The objective of this work is to investigate which forcing affects the fluctuation structure of groundwater levels
in Wallingford. Hence, the recharge-groundwater flow model, with the selected optimum parameter set, is used
to simulate groundwater levels while varying the input time series and parameters as explained below. The
selected inputs and parameters, which will be varied, are highlighted in Figure 3 in orange and the fractal
behaviour of the simulated groundwater levels will be analysed using r-DFAn.
The procedure adapted for varying the parameters and input series is as follows: the selected optimum parameter
values will be rescaled within certain limits that are found to produce reasonable groundwater levels in both time
and fractal domains, a random permutation of river stage will be used to test the effect of river stage's fractal
behaviour on that of groundwater levels because randomly shuffling the series will break its temporal structure,
and finally, rainfall input series with different fractal properties will be simulated and used to drive the coupled
recharge-groundwater flow model.
The rainfall model used to generate rainfall realizations with different fractal behaviour is explained in the
following section.

**2.4 Stochastic Rainfall Model**
The $\beta$-lognormal model used in (Molnar & Burlando, 2008; Over & Gupta, 1994; Paschalis, Molnar & Burlando,
2012), which is a discrete multiplicative random cascade, will be used to downscale different realisations of the
observed rainfall series. This is done by aggregating observed series to a daily time scale and then using the
cascade generator for downscaling. The cascade generator ($w$) is described as follows (Over, 1995):
$$w = w_\beta w_{log\,n} \tag{1}$$

where $w_\beta$ is the $\beta$ model's cascade generator and $w_{log\,n}$ is the lognormally distributed cascade generator of the
lognormal model and both are computed as follows (Over & Gupta, 1994; Over, 1995):
$$w_\beta = \begin{cases} 0 \; with \; probability \; p = 1 - 2^{-\beta} \\ 2^\beta \; with \; probability \; 1 - p = 2^{-\beta} \end{cases} \tag{2}$$

$$w_{log\,n} = 2^{\mu + \sigma X} \tag{3}$$

where $\mu$ and $\sigma$ are, respectively, the mean and variance of the lognormal cascade generator ($w_{log\,n}$) and $X$ is a
standard Gaussian random variable. To preserve the mean of the generated rainfall series when downscaling, $\mu$





and $\sigma$ are not independent. $\beta$ and $\sigma$ are essential for describing the scaling field of the rainfall series, and it is from
observed rainfall's scaling field that the parameters are calibrated (Molnar & Burlando, 2008). The $\beta$ parameter
indicates the level of intermittency of the generated rainfall series (Molnar & Burlando, 2008).
In this context, the performance of the rainfall model is assessed based on its ability to preserve observed rainfall's
basic statistical properties such as its mean, standard deviation, probability of dry periods and its distribution. The
assessment of the model's performance is performed at a number of aggregation scales as shown in Figure 6. The
performance of the model was found satisfactory for simulating rainfall at Wallingford. It should be noted that
due to the discrete nature of the multiplicative random cascade, there is an overestimation of the probability of
no-rainfall at the daily scale. The reason is that there is a non-zero probability that the downscaled rainfall is zero
(e.g. if the first 2 multiplicative weights $w_\beta$ are both zero), even if the rainfall depth at the daily scale where the
downscaling procedure started, is not.

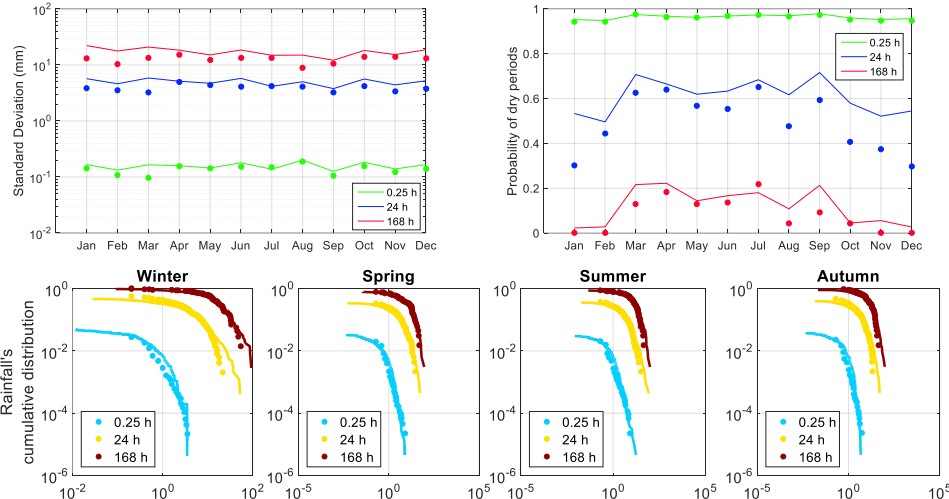

**Figure 6. Top left: comparison between the standard deviations of each month of observed data (dots) and simulated**
**rainfall (lines). Top right: comparison between the probability of dry periods of each month of observed data (dots)**
**and simulated rainfall (lines). Bottom: comparison between empirically fitted cumulative distributions of observed**
**data (dots) and simulated rainfall (lines) for each season. Three scales are selected for the model's performance**
**assessment: 15 minutes, 1 day and 1 week**
Following the calibration, the stochastic rainfall model is used to simulate various rainfall series of different fractal
behaviour, and this is done by changing the values of the calibrated parameters. Results of this exercise, in addition
to other results, are presented in the following section.

**3 Results and Discussion**
Time series and parameters used to drive the coupled recharge-groundwater flow model are altered to investigate
their impact on the fractal behaviour of the simulated groundwater levels. The time series/parameters are changed
one-at-a-time (while keeping the remaining time series/parameters unchanged) and are used to drive the model.
This implementation will show which local forcing changes the groundwater level's fractal behaviour. In other
words, this is a sort of sensitivity analysis of the fractal behaviour of the simulated groundwater levels, however,
the simulation is performed in the time domain and using a physically based model which will help us relate
changes in the fractal behaviour to physical phenomenon.
The effect of the following on groundwater levels' fractal behaviour in the Wallingford study site is be studied:
-    Rainfall's fractal behaviour.
-    Aquifer's physical properties. These include the hydraulic conductivity and specific yield of the shallow
198        unconfined aquifer in Wallingford.
-    Empirical parameters in the Van-Genuchten-Mualem Model describing the water retention curve.
-    River stage's fractal behaviour and its distance from the borehole at which GWLs are observed.





Based on the results, we have divided the above time series/parameters into two main categories: sensitive ones
and non-sensitive ones. Sensitive factors are those that produce statistically different fractal behaviour in
groundwater levels and vice versa are the non-sensitive factors. Rainfall fractal behaviour resulted in statistically
significant change in the groundwater levels' global fractal behaviour. The remaining factors did not, however,
some were found to affect groundwater level's fractal behaviour on larger scales and others affected smaller scales
as will be discussed in the following sections.

**3.1 Sensitive Factors**
**3.1.1 Rainfall**
Using the stochastic rainfall model, different values of the intermittency parameter $\beta$ are used to simulate rainfall
series of varying fractal properties. By altering the $\beta$ parameter we focus on the scaling of the probability of zero
rainfall. For every change in the $\beta$ parameter, 5 realisations are simulated. A total of 40 rainfall realisations were
simulated with resulting global scaling behaviour ranging from 0.6 to 1.05. Figure 7 presents a number of
simulated groundwater level series using simulated rainfall. The range of $\beta$ parameter was large enough such that
intensities at the lowest scale between extreme case simulations differ significantly. Rainfall amount at the daily
scale are, for all simulations, preserved on average.

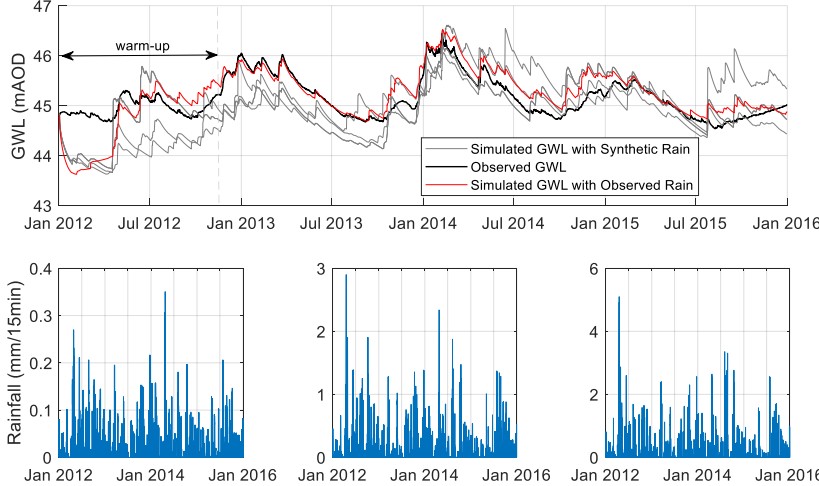

**Figure 7. Top panel: groundwater levels simulated using observed rainfall and selected rainfall realisations. Bottom**
**panels: selected rainfall realisations.**
The 40 rainfall realisations were used to drive the coupled recharge-groundwater flow model to simulate 40
drainage series and 40 groundwater levels. Figure 8 presents a summary of the global scaling exponents of all
simulated rainfall realisations and corresponding drainage and groundwater levels. Notable is that the rainfall
realizations for different $\beta$ parameters have statistically different global scaling exponents and these in turn
produce statistically different global scaling exponents for both drainage and groundwater levels. Additionally,
simulations with $\beta \times 1.0$, i.e. with no change to the calibrated values, result in values of global scaling exponents
that are similar to the observed values (Figure 8).



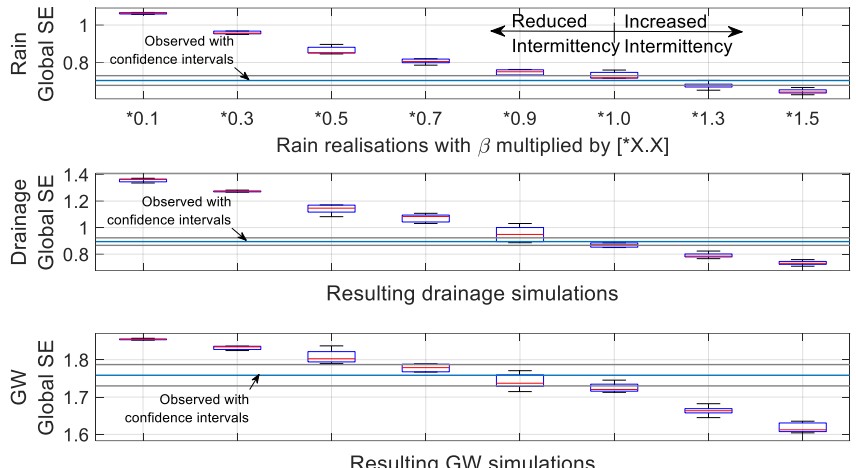

**Figure 8. Box plots summarising the global scaling exponents of the 40 simulated rainfall realisations (top panel) and corresponding drainage (middle panel) and groundwater levels (bottom panel). The red line represents the median, box edges represent the 25th and 75th percentiles, whiskers represent the maximum range.**

Looking further into the effect of rainfall's fractal behaviour, we find that changes to rainfall's global scaling exponent strongly affects the fractal behaviour of drainage. This is evident from Figure 9 when comparing the slopes which describe the change of global scaling exponent of each variable relative to changes in rainfall's global scaling exponent, where changes in the global scaling exponents of drainage are significantly larger than those of both rainfall and groundwater levels.

This was attributed to the unsaturated zone which magnifies the effect of extended dry periods in the case of an intermittent rainfall signal or wetter circumstances in the case of a less intermittent rainfall signal. Additionally, the relatively wider range of variation of global fractal behaviour in the recharge signal was narrowed down as recharge flows into the saturated zone to produce groundwater fluctuation.

This illustrates how groundwater is isolated from atmospheric changes to a great degree by the unsaturated zone and it takes a magnified change to the recharge signal to produce statistically significant change to the fluctuation structure of groundwater.

A novel finding is the effect of change of global fractal behaviour of rainfall series on that of drainage/recharge and groundwater levels. Previous publications have highlighted the increase in memory of a white noise or observed rainfall series as it infiltrates through soil (Gelhar, 1974; Yang, Zhang & Liang, 2017; Zhang & Schilling, 2004), however, comparing the degree of change of global fractal behaviour between rainfall, drainage and groundwater levels has, to the best of our knowledge, not been investigated previously.



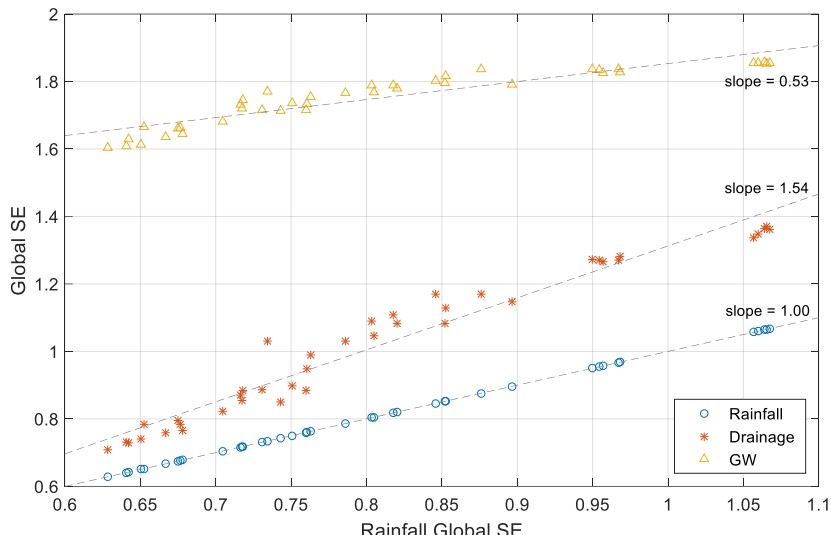

**Figure 9. Scatter plot of simulated rainfall, drainage and groundwater levels' global fractal behaviour vs. simulated rainfall's global fractal behaviour**

Further investigation of the effect of rainfall on both drainage and groundwater levels was performed by studying the effect of *local* fractal behaviour of rainfall on that of drainage and groundwater levels. This was done by investigating the degree of correlation between rainfall's local fractal behaviour and that of both drainage and groundwater levels.

Local fractal behaviour is described in terms of local scaling exponents and crossovers. Relating crossover locations is difficult because the number of crossovers is seldom equal in the series being compared and hence crossovers in two series cannot always be associated with each other. Local scaling exponents extend over different ranges of scales, and hence, comparing local scaling exponents is not straight forward either. This is illustrated in the left panel of Figure 10 where neither crossovers nor local scaling exponents in series A can be individually associated to those in series B. Hence, as illustrated in Figure 10, the r-DFA plot is transformed into a different series that contains information about the local scaling exponent and the range of scales over which it extends (hence indirectly reflecting the crossover location). The correlation coefficient of the transformed series is then determined and the results of the 40 rainfall realisations and its corresponding drainage and groundwater levels are presented in Figure 11. The correlations between rainfall and drainage, rainfall and groundwater levels, and drainage and groundwater levels are determined.



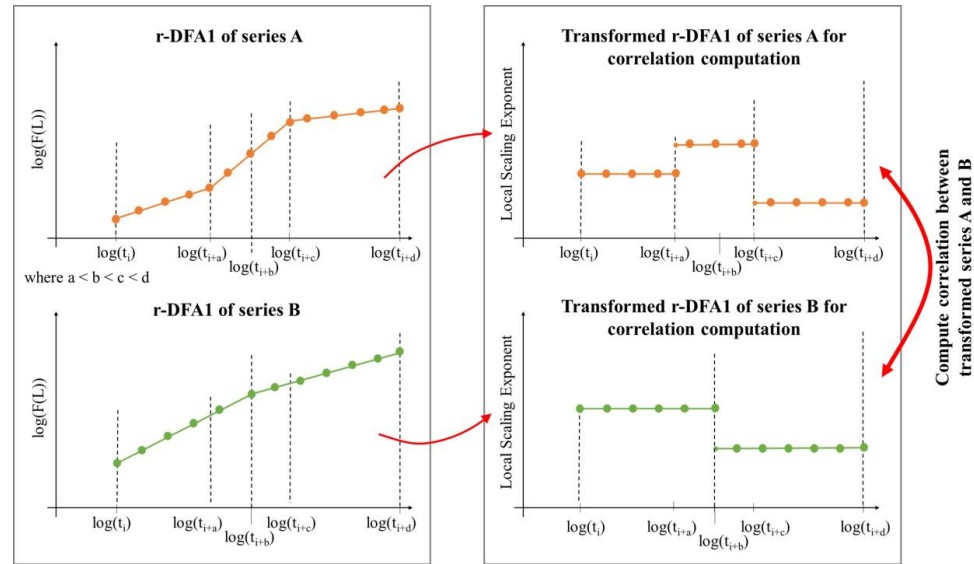

**Figure 10. Illustration that explains how the r-DFA1 plots are transformed in order to be able to compute a**
**correlation coefficient between pairs of r-DFA1 plots.**
60%, 70% and 80% of the correlation coefficients determined between, respectively, rainfall and drainage, rainfall
and groundwater levels, and drainage and groundwater levels, are higher than 0.7 (Figure 11). The bottom
illustration in Figure 11 summarizes the correlation coefficients determined. They all lie towards the higher end
of correlations with the correlations between drainage and groundwater levels significantly different than the other
two correlation groups (evident from the non-overlapping confidence intervals).



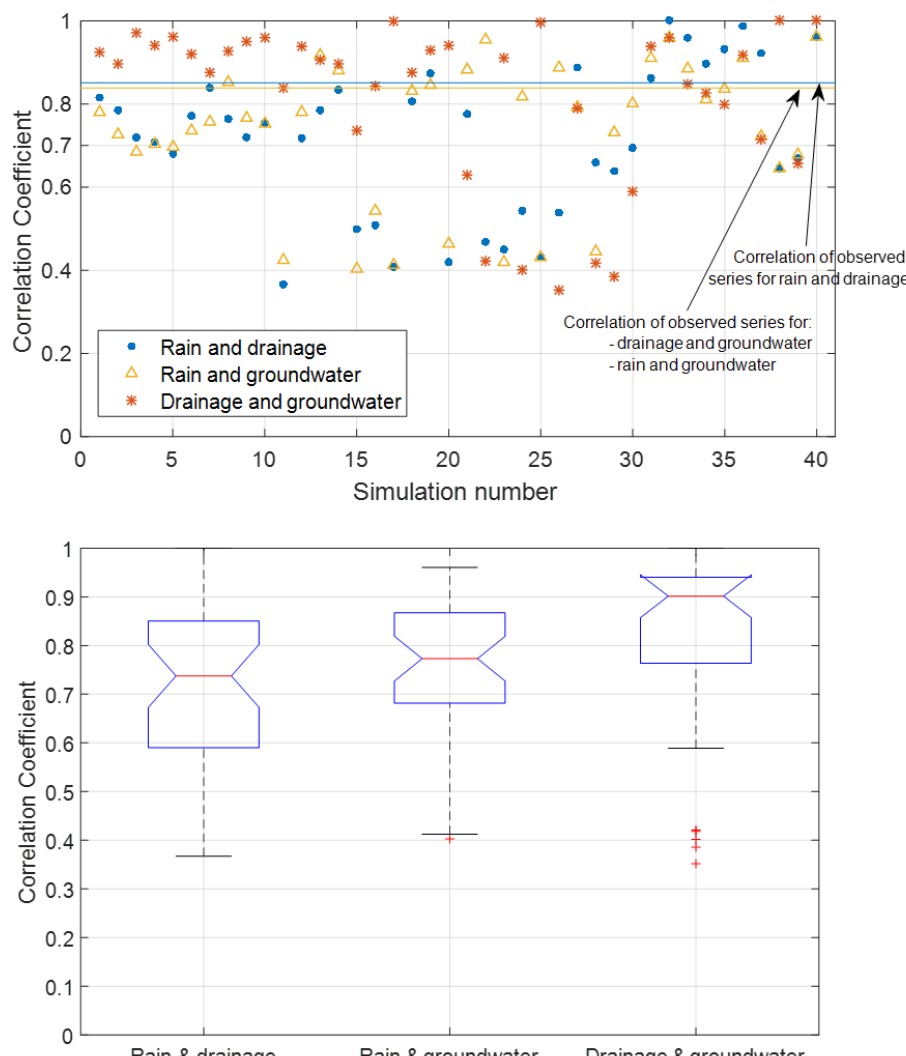

**Figure 11. Top panel: correlations between the 40 realisations of rainfall, drainage and groundwater levels' local scaling exponents (r-DFA1). Bottom panel: Boxplots summarising the results presented in the top panel. The red line represents the median, notches represent the confidence intervals of the median with 95% significance level, box edges represent the 25th and 75th percentiles, red crosses represent outliers, and, whiskers represent the maximum range excluding the outliers.**

This shows preliminary evidence that local fractal behaviour in rainfall may affect local fractal behaviour in both drainage and groundwater levels. Nevertheless, this should be investigated further in other locations or using different models because of the non-negligible number of realisations that are not strongly correlated.



### 3.2 Non-Sensitive Factors

#### 3.2.1 Hydraulic Conductivity and Specific Yield

Contrary to speculations of the dependence of groundwater's fractal behaviour on the aquifer's physical properties (Li & Zhang, 2007; Yu et al., 2016; Zhang & Schilling, 2004), results from the Wallingford site using the recharge-groundwater flow model used here shows that changes to the physical parameters – the hydraulic conductivity and specific yield – between a range of 50% and 500% does not produce differences in the global fractal behaviour of groundwater levels that is statistically significant (top panels in Figure 12). As mentioned before, changes are made to one parameter at a time and the optimised parameter value is used as the starting point.

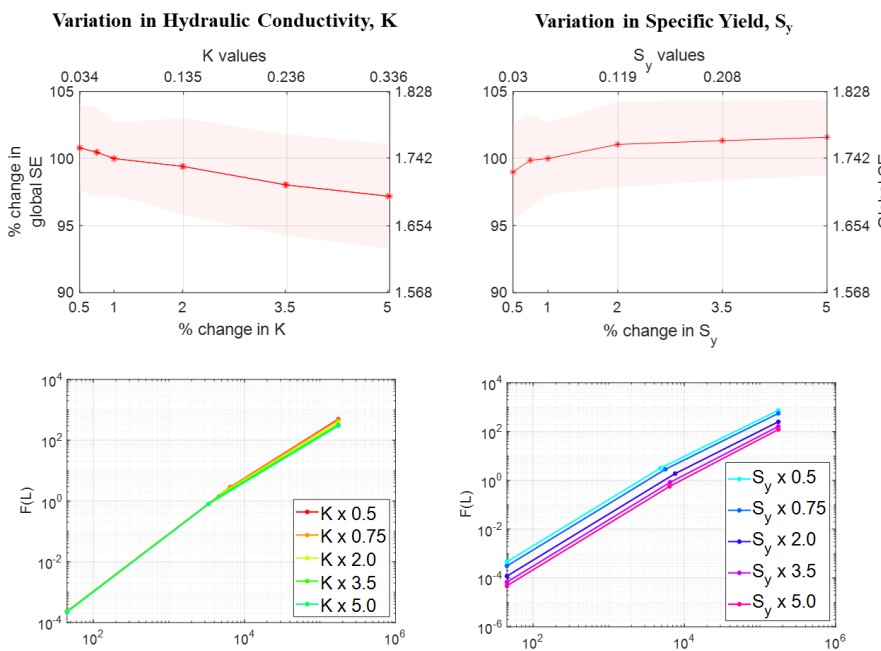

**Figure 12. Top panels: Effect of change of hydraulic conductivity (left) and specific yield (right) on the global scaling exponent of simulated groundwater level with 95% confidence intervals. Bottom panels: r-DFA1 results of groundwater levels simulated with varying hydraulic conductivities and specific yield.**

The bottom panels in Figure 12 present the r-DFA1 plots for the various hydraulic conductivity and specific yield values used. Even though there is no significant change to the global fractal behaviour of groundwater levels, one observes that, for changes to hydraulic conductivity (bottom left panel in Figure 12), changes tend to occur on larger scales (greater than a number of days), and with changes to the specific yield (bottom right panel in Figure 12), there is a general reduction in groundwater level's variance with increase in specific yield, however this change is constant over all scales because there is very minor change to the groundwater level's global fractal behaviour but there is a reduction in the mean of the variances of the r-DFA1 plots (as shown in the bottom right panel in Figure 12).

#### 3.2.2 Recharge Parameters

The same procedure followed for the aquifer's physical parameters, the recharge parameters were varied between 25% and 175% of the optimized values. This range was found to produce groundwater levels that are acceptable given the aquifer's dimensions and the river levels. The recharge parameters ($m$ and $\eta$) are empirical parameters from the Van Genuchten-Maulem Model used in the SMAP recharge model (Habib, Abrar et al., 2022).





Figure 13 (top left and right) shows the overlapping confidence intervals of the global scaling exponents of
groundwater levels that are simulated using different recharge parameters. The $m$ recharge parameter affects
smaller scales (smaller than days) in the groundwater levels scaling behaviour (bottom left panel in Figure 13),
contrary to the effect of hydraulic conductivity which affects the larger scales only. The $\eta$ parameter does not
appear to affect groundwater levels local fractal behaviour in any way (bottom right panel Figure 13). The effect
that recharge has on the smaller scales of groundwater levels' fractal behaviour can be related to previous work
(Katul et al., 2007).

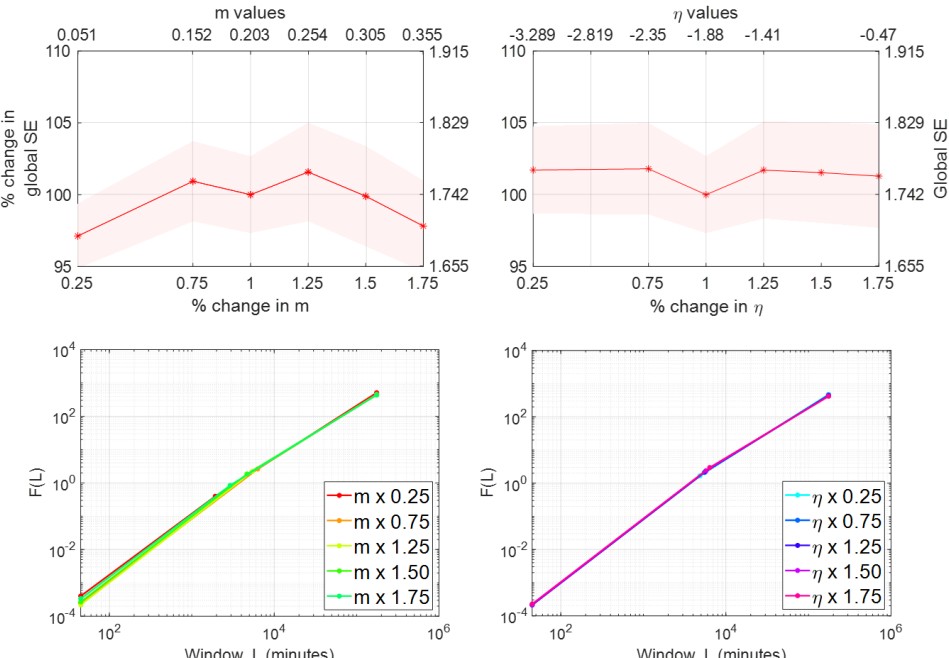

**Figure 13. Top panel: Effect of change of different recharge parameter values on the global scaling exponent of**
**simulated groundwater level with 95% confidence intervals. Bottom: r-DFA1 results of groundwater levels simulated**
**with varying recharge parameter values.**

**3.2.3 River Stage's Fractal Behavior and Distance from Groundwater Level Measurements**
Simulating groundwater levels with the observed river stage series after randomly shuffling it (i.e. after breaking
its scaling structure while maintaining the original distribution of the series) did not affect groundwater levels that
are monitored at a distance of 420m from the river. This illustrates that the fractal behaviour of river stage does
not affect groundwater levels at this distance.
Ground water levels closer to the river, at a vicinity of 100m, on the other hand, showed small change to the global
fractal behaviour with change to the river stage's fractal behaviour (Figure 14 middle panel). Notable, as well, is
the reduction of groundwater level's fractal behaviour to values lower than that of river stage's global fractal
behaviour. This is explained by the fact that the flow of groundwater computed by the recharge-groundwater flow
model is governed solely by change in head gradient (Darcy's Equation) and the complex dynamics at the river-
aquifer interface are not modelled. Hence, at close vicinity to the river, fluctuation of river stage may result in
reverse flows (i.e. flow from the river into the aquifer) during some periods as shown in the simulated time series
in the top panel of Figure 14, which, may not be the case in reality. It is speculated that in reality the global fractal
behaviour of groundwater levels is not lower than that of river stage (Little & Bloomfield, 2010). However, in
order to ascertain the correctness of this hypothesis, observations of groundwater levels closer to the river should
be analysed.

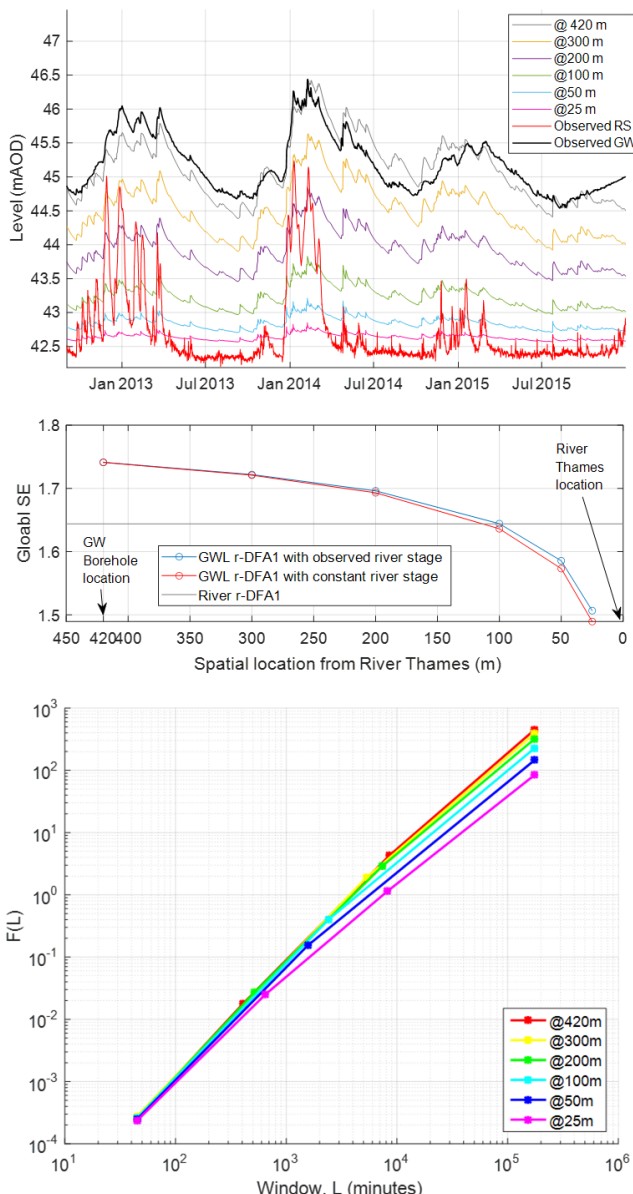

**Figure 14. Top panel: Simulated groundwater time series at different locations. Middle Panel: Global fractal**
**behaviour of simulated groundwater levels at different locations with, first, observed river stage as boundary**
**condition and then a constant (mean value) river stage as boundary condition. Bottom panel: Local scaling behaviour**
**of first order (i.e., r-DFA1) at different locations.**

Figure 14 also shows that the groundwater level's local fractal behaviour is affected mainly across larger scales
especially for groundwater levels closer to the river and this is similar to previously published results (Liang,
Xiuyu & Zhang, 2013).




**4 Summary, Conclusions and Recommendations**

A physically-based recharge-groundwater flow model, that was developed, calibrated and assessed in both time
and fractal domains for a riparian aquifer in Wallingford, United Kingdom (Habib, Abrar et al., 2022), has been
used here to study the sensitivity of groundwater levels' fractal behaviour to various forcings and parameters
required to run the model. The forcings and parameters considered were rainfall's fractal behaviour, the aquifer's
physical parameters, the empirical parameters for simulating recharge, the river stage's fractal behaviour and its
distance from the borehole at which groundwater levels were measured.
It was found that changes to rainfall's fractal behaviour – which were simulated by changing a parameter that
represents rainfall's intermittency in a stochastic rainfall model, – was the only factor that resulted in statistically
different global fractal behaviour of groundwater levels. Furthermore, the local fractal behaviour of rainfall was
found to influence the fractal behaviour of recharge and groundwater levels. While this paper presents evidence
that the local fractal behaviour of rainfall is indeed transferred to drainage and then to groundwater levels, further
investigation of this is required.
With the help of a reliable method for studying fractal behaviour, which, in this case, was robust detrended
fluctuation analysis (r-DFAn), our perception of the factors that influence the fluctuation structure of a time series
is improved and this was illustrated. Nevertheless, repeating this exercise using different hydrological models and
for different sites is essential for confirming the results found.
Additionally, the issue of parameter interaction during calibration/optimization may be projected on this study
where certain combinations of change to parameters may yield significant change to groundwater level's fractal
behaviour. This may be noticed when observing the change that the recharge parameters and the aquifer
parameters had on the local fractal behaviour of groundwater levels where the former affected smaller time scales
and the latter affected larger time scales.

**Competing Interests**

The authors declare that they have no conflict of interest.

**Acknowledgments**

Mr James Sorensen and Dr John P Bloomfield publish with the permission of the Executive Director of the
British Geological Survey (NERC).
We extend our gratitude to Ms Katie Muchan from the Centre for Ecology and Hydrology, Wallingford, United
Kingdom, for making available for this work the meteorological data monitored at the Wallingford site.
The meteorological data can be requested from UK Centre for Ecology and Hydrology (UKCEH) from the
following link: https://www.ceh.ac.uk/our-science/projects/wallingford-met-site and the river stage and
groundwater level data, which are managed by the British Geological Survey (BGS), can be downloaded from
10.5285/637eeed6-7175-4346-9321-0c14332456c6 (Sorensen, 2022)

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

groundwater recharge. *Water Resources Research.* 40 (3), .





## Appendix A

**Table A. 1 A non-exhaustive list of published research that involves the study of fractal behaviour along with the use of models.**

| Paper | Model used | Variable analysed | Fractal analysis method | Summary/ relevant outcomes/ relevant highlights |
|---|---|---|---|---|
| (Gelhar, 1974) | • Linear Reservoir Model, <br>• Dupuit Aquifer Model, and <br>• Laplace Aquifer Model, all represented in the spectral domain | Groundwater levels | Stochastic Spectral Analysis | The analytical models were found to properly replicate the spectral behaviour of the groundwater system when the models were properly calibrated. Hence, they suggested the use of spectral analysis to determine the aquifer's parameters. |
| (Duffy, C. & Gelhar, 1986; Duffy, C. J. & Gelhar, 1985) | Three transport models expressed in the frequency domain which are: <br>• Lumped parameter linear reservoir model, <br>• convective (advective) dispersion in a curvilinear flow field, and <br>• convective-dispersive transport in a uniform flow field | Solute transport in groundwater | Power spectral analysis | Parameters of the physical system are determined in the frequency domain by comparing theoretical and observed spectral response and using 'type curve techniques'. Based on the type of contaminant source and groundwater flow fields (i.e. uniform or non-uniform), unique spectral behaviours are observed. |
| (Zhang & Schilling, 2004) | Linear reservoir model (in spectral domain) that was used in (Gelhar, 1974) | Recharge | Power spectral analysis | The recharge signal, estimated from the model, exhibited scaling and the value of the scaling was found to be dependent on the specific yield and transmissivity of the aquifer (based on the theoretical model used). |
| (Zhang & Li, 2006) | • Numerical simulation of Boussinesq Equation and <br>• spectral representation of the linear reservoir model used in (Gelhar, 1974) | Groundwater levels | Power spectral analysis | Recharge with known spectral properties was simulated using derived equations for covariance and variance from the linear reservoir model. Spectral properties of groundwater levels simulated using the Boussinesq equation and the simulated recharge as input matched those determined using the linear reservoir model. |





| Paper | Model used | Variable analysed | Fractal analysis method | Summary/ relevant outcomes/ relevant highlights |
|---|---|---|---|---|
| (Katul et al., 2007) | Spectral model derived from the water balance equation that determines soil-moisture's memory | Soil moisture | Power spectral analysis | Using the analytical model with white noise precipitation as input, the resulting soil moisture exhibits memory at the shorter time scales (higher frequencies) and is a white noise at larger time scales. Precipitation is believed to govern the soil moisture memory at the shorter time scales (higher frequencies). There is energy imbalance in the measured soil moisture series and this implies that for time scales greater than 12 hours, the diurnal cycle in evapotranspiration can be ignored. |
| (Lo & Famiglietti, 2010) | National Centre for Atmospheric Research Community Land Model (a land surface model) | Soil moisture | Power spectral analysis | Spectral analysis was used to study the effect of including a groundwater module in a Land Surface Model. They concluded that the land surface hydrologic memory, estimated from soil moisture, is dependent on the depth of groundwater levels. |
| (Thompson & Katul, 2011) | Some of the models used: <br> • Deterministic models: Linear catchment water balance, non-linear water balance (such as Boussinesq Equation) <br> • Stochastic models: reservoirs in parallel/series with random time constants. | Streamflow | Power spectral analysis | Classic linear systems replicated the observed streamflow power spectra well. |
| (Istanbulluoglu et al., 2012) | (Annual) linear reservoir model coupled with the Budyko hypothesis | Runoff, groundwater dynamics | Hurst coefficient | Aquifer water storage and the aridity index, along with the stochastic nature of the input climate series, are believed to be the governing factors for the effect that climate series have on transforming precipitation to groundwater. |
| (Russian et al., 2013) | A multicontinuum approach which is an extension of the classical | Aquifer discharge | Power spectral analysis | The approach presented relates the scaling of the frequency transfer function with the aquifer's storativity, catchment size and a stochastic representation of |



| Paper | Model used | Variable analysed | Fractal analysis method | Summary/ relevant outcomes/ relevant highlights |
|---|---|---|---|---|
| | Linear and Dupuit Models | | | heterogeneity of hydraulic conductivity. |
| (Liang, Xiuyu & Zhang, 2013) | Boussinesq Equation represented in spectral form | Groundwater levels | Power spectral analysis | The analytical representation of groundwater spectral behaviour can be fitted to observed groundwater spectra, hence, the parameters of the analytical expression can be fitted using observed data. Scaling of groundwater levels are found to be affected at longer time scales by the existence of a constant head boundary which results in a crossover. |
| (Condon & Maxwell, 2014) | Integrated physical hydrology model ParFlow-CLM | Groundwater fluctuation in irrigated catchments and latent heat flux. | Power spectral analysis | Irrigation affects the temporal behaviour of groundwater levels. The idea of a 'fractal filter' is demonstrated. Water table fluctuations appear to be affected by differences in hydraulic conductivity. Water management operations (such as pumping and irrigation) seem to add persistence to the groundwater levels. |
| (Williams & Pelletier, 2015) | Linear Langevin Equation (the Bousinesq Equation with a white noise recharge input) | Lake-level fluctuation | Power spectral analysis | The model reproduced the size-dependent spectral scaling of lake-levels. |
| (Rahman, Sulis & Kollet, 2016) | ParFlow and common land model (ParFlow.CLM) | Soil moisture, evapotranspiration, and other land surface processes | Continuous Wavelet Transform | From model simulations, groundwater dynamics are found to affect the variance of land surface processes and potentially the forecast of hydrological droughts. |
| (Liang, X., Zhang & Schilling, 2016) | Boussinesq Equation represented in spectral form | Groundwater levels | Power spectral analysis | Heterogeneity of the aquifer's transmissivity increases the variation of groundwater levels. |
| (Yang, Zhang & Liang, 2017) | GSFLOW which combines USGS's precipitation-runoff modelling system (PRMS) with MODFLOW-2005 | Precipitation, infiltration at the land surface, seepage through unsaturated zone, recharge to water table, groundwater flow and discharge from aquifer. | Power spectral analysis | The hydrological system acts as a cascade of hierarchical fractal filters which transforms white noise to a fractal signal. The unsaturated zone exhibits the greatest dampening effect compared to the land surface and unsaturated zone. Simulated soil moisture series has increased temporal scaling at increased vertical depth. |
| (Habib, Abrar et al., 2022) | Coupled recharge-groundwater flow model. The models include a soil moisture accounting | Groundwater levels | Robust detrended fluctuation analysis, r-DFAn (Habib, A. et al., 2017) | The physically-based model replicated the groundwater level's fractal behaviour to an acceptable degree. The concept of 'fractal-domain-refinement' was introduced and this involves using fractal |



| Paper | Model used | Variable analysed | Fractal analysis method | Summary/ relevant outcomes/ relevant highlights |
|---|---|---|---|---|
| | procedure (Mathias et al., 2015) and a 1D Boussinesq Equation model. | | | behaviour of the simulated variable to refine the optimum parameters determined through optimisation. |