# Peer review of "Study of the effect of local forcing on the fractal behavior of 1 shallow groundwater levels in a riparian aquifer 2"

_Hydrology and Earth System Sciences, 2023_

## Author Comment (AC1)

Dear Reviewer,

Thank you for taking the time to review our manuscript and provide feedback. After reviewing the comments, we prepared the following response in tabular form.

| | Reviewer's Comments | Authors' Responses |
|---|---|---|
| 1 | The manuscript includes material that can be seen as a direct continuation of a series of works by some of the lead co-Authors (as also apparent from the reference list). It is also focused on the very same experimental site tackled in previous works and relies on the same data (or mostly). While being technically correct, a first critical point associated with the current work is its incremental value with respect to previous published work by the group. Analysis of the fractal nature of quantities associated with the groundwater flow simulations are illustrated in prior works, albeit not directly related to the fractal nature of rainfall. All in all, the results illustrated in the current contribution are seen as a straightforward extension of the previous work, in this sense. Additionally, they are illustrated with only minimal insight on physics underpinning the documented results. | Indeed, the manuscript is a continuation of work that the authors have worked on, however, we beg to differ when it comes to its novelty and contribution. The two main novelties of this work are: the first is the use of a robust detrended fluctuation algorithm to objectively quantify fractal behaviour which helps determine statistically significant changes to fractal behaviour (section 2.2), and the second is the use of a multiplicative random cascade rainfall model to simulate rainfall series with different fractal behaviour after calibrating it to the observed rainfall (section 2.4).

As for its contribution, thanks to the rigorous sensitivity analysis we performed, we were able to infer insights into how the fractal behaviour is transferred from rainfall through the unsaturated zone and finally transferred to groundwater levels. The fractal behaviour in the unsaturated zone is surprisingly magnified (figure 9) and this opens the doors to further investigating how fractal behaviour arises in infiltration in the unsaturated zone. |
| 2 | With reference to the data, I am not sure the uncertainty associated with these has been explored. This might be considered as an issue which is not too critical at this point. | To minimise uncertainty and provide confidence in the results the data have had thorough quality control checks, as described in detail in: https://www.sciencedirect.com/science/article/abs/pii/S0022169417302548. This is why we agree with the reviewer that this is not a critical issue for the purposes of this work. |

| | | Reviewer's Comments | Authors' Responses |
|---|---|---|---|
| 3 | | Additional comments include the lack of a modern sensitivity analysis (either Local or Global) from which one can see clear contributions of model parameters to the uncertainty of model outputs and their importance to it in a quantifiable manner. | Sensitivity analysis and optimization of the sensitive parameters are performed. They are described in section 2.3 and Figure 3. |
| 4 | | The assumption of homogeneity of the subsurface system seem to be too limiting to discern the impact of, e.g., hydraulic conductivity on the evidenced multifractal behavior of groundwater levels. I would have suggested enhancing the possibility of discerning such an impact upon relying on a randomly heterogeneous distribution of conductivity, for instance. | To benefit from the high-resolution data and the simplicity of the study site, our approach involved using the simplest possible physically based model to successfully simulate the fractal behaviour of groundwater levels based on our knowledge of the geology and nature of the study site. We could not justify any increase in the complexity of the model. The geology of the riparian aquifer is known to be relatively uniform and hence treated as being homogenous. This has been discussed in a previous work: https://www.sciencedirect.com/science/article/abs/pii/S0022169417302548 |
| 5 | | Additionally, it is noted that the Authors rely on a previously calibrated model and employ the estimated model parameters as a guidance around which they then vary them in their simulations. The interval of variability of the model parameters around their estimated counterparts should be driven by the estimation uncertainty associated with the inverse modeling results. These are not reported (and I was not able to find these in previous material). As such, it is not clear how the variability of model parameters is constrained to the available data. | Each parameter was varied in accordance with two factors:
- The value of the physical parameter is reasonable, for example, hydraulic conductivity is consistent with the range of values typically reported in the literature.
- Variation of empirical parameters was constrained by the requirement that simulated groundwater levels in the time domain were visually acceptable. This is mentioned in the first paragraph in section 3.2.2.

Nevertheless, indeed this is not clearly explained in the manuscript. We will edit the manuscript to clarify this. |